# Adverse Events Comparison of Double Beta-Lactam Combinations for Bloodstream Infections: Ampicillin plus Ceftriaxone and Ampicillin/Cloxacillin

**DOI:** 10.3390/antibiotics13080696

**Published:** 2024-07-25

**Authors:** Kazuhiro Ishikawa, Daiki Kobayashi, Nobuyoshi Mori

**Affiliations:** 1Department of Infectious Diseases, St. Luke’s International University, Tokyo 104-8560, Japan; morinob@luke.ac.jp; 2Department of Primary Care and General Medicine, Tokyo Medical University Ibaraki Medical Center, Ibaraki 300-0332, Japan; daikik0503@gmail.com

**Keywords:** acute kidney injury, ampicillin/cloxacillin, beta-lactam, double beta-lactam

## Abstract

In Japan, only ampicillin/cloxacillin (ABPC/MCIPC) is available as an anti-staphylococcal penicillin-based treatment for *Staphylococcus aureus* bacteremia. However, the incidence of adverse events associated with double beta-lactam administration remains unknown. Therefore, we investigated the adverse events of double beta-lactam administration in patients with bacteremia. Adult patients (≥18 years) with bacteremia treated with ABPC, ABPC + ceftriaxone (CTRX), or ABPC/MCIPC were retrospectively analyzed. The primary outcome of this study was the incidence of adverse events such as acute kidney injury, liver dysfunction, and myelosuppression. Chi-square tests and *t*-tests were used for bivariate analysis. Propensity score (PS) matching was conducted to adjust for confounding factors. We included 277 ABPC-, 57 ABPC + CTRX-, and 43 ABPC/MCIPC-treated patients. Significant differences were noted in age, number of male patients, proportion of patients with qSOFA score ≥2, incidence of chronic kidney disease, treatment duration, mechanical ventilation use, vasopressor use, and proportion of patients with acute kidney injury (AKI) KDIGO grade ≥2. Further, a significant difference was observed between ABPC and ABPC/MCIPC, with a hazard ratio of 1.83 in AKI. In the PS-matched cohort, AKI incidence associated with ABPC/MCIPC was significantly higher than that associated with ABPC. ABPC + CTRX may be safe, whereas ABPC/MCIPC presents a higher risk of AKI and may not be suitable.

## 1. Introduction

Commonly used antimicrobials in clinical practice belong to the class of beta-lactam antimicrobials, including penicillin and cephalosporins. Potential adverse events associated with these beta-lactam antimicrobials can manifest as allergic reactions such as skin rashes and renal injuries, including acute interstitial nephritis, acute tubular necrosis, liver dysfunction, and myelosuppression, which lead to conditions such as leukopenia, thrombocytopenia, or anemia [1].

Standard treatments for methicillin-sensitive *Staphylococcus aureus* (MSSA) bacteremia involve anti-Staphylococcus penicillins (ASPs) such as oxacillin, nafcillin, and cloxacillin (MCIPC). In Japan, only the combined formulation of ampicillin (ABPC)/MCIPC is commercially available, necessitating combination therapy with beta-lactam antimicrobials. However, ABPC is ineffective against penicillinase-producing MSSA. Cefazolin (CEZ) is frequently used as an alternative therapy for MSSA bacteremia. Meta-analysis has revealed that CEZ therapy significantly reduces mortality in MSSA bacteremia cases compared to that observed when treated with ASPs without affecting clinical failure rates and is found to be better tolerated [2]. However, another meta-analysis indicated a relatively higher risk of reinfection with MSSA when using CEZ [3], and ASPs continue to be used owing to the absence of randomized trials comparing CEZ with ASPs.

Beta-lactam combination therapy may represent a viable option in the following cases: infective endocarditis caused by *Enterococcus faecalis*, where ABPC plus ceftriaxone (CTRX) shows a synergistic effect; bacterial meningitis, where ABPC is effective against *Listeria monocytogenes* and CTRX is effective against *Streptococcus pneumoniae* and *Neisseria meningitides;* and urinary tract infections potentially caused by *Streptococcus* spp. and Enterobacteriaceae such as *Escherichia coli*, which is highly resistant to ABPC/sulbactam (SBT) but susceptible to the considerably broader spectrum piperacillin/tazobactam.

Antimicrobial-resistant strains have developed owing to the broad-spectrum use of antimicrobials. The use of double beta-lactams may reduce the reliance on broad-spectrum beta-lactam/beta-lactamase inhibitors and carbapenems; however, the incidence rate of adverse events remains uncertain.

Here, we aimed to compare the incidence of adverse events, such as acute renal injury, liver dysfunction, and myelosuppression, between patients treated with ABPC, ABPC + CTRX, and ABPC/MCIPC. Given the variation in disease severity associated with different antimicrobial groups, our study focused on patients with bacteremia. Our findings suggest that ABPC + CTRX treatment may be safe under certain conditions, whereas ABPC/MCIPC treatment may present a higher risk of developing acute kidney injury (AKI) and not be suitable for treating bacterial infections.

## 2. Results

The baseline characteristics of the patients included in this study are shown in Table 1. We included 277 patients who received ABPC, 57 who received ABPC + CTRX, and 43 who received ABPC/MCIPC (Appendix A). Significant differences were found in the following baseline characteristics: average age (71.1 years [standard deviation (SD) of 16.5 years] for ABPC, 64.0 years [16.9 years] for ABPC + CTRX, and 56.6 years [17.5 years] for ABPC/MCIPC), proportion of male patients (62.5% for ABPC, 47.4% for ABPC + CTRX, and 72.1% for ABPC/MCIPC), incidence of chronic kidney disease (39.0% for ABPC, 35.1% for ABPC + CTRX, and 18.6% for ABPC/MCIPC), and the proportion of patients with a quick sepsis-related organ failure assessment (qSOFA) score ≥2 (23.1% for ABPC, 64.4% for ABPC + CTRX, and 41.9% for ABPC/MCIPC). Vancomycin was administered for more than 3 days in a subset of each group (7.60% for ABPC, 28.1% for ABPC + CTRX, and 41.9% for ABPC/MCIPC). Hepatic dysfunction on admission was indicated based on total bilirubin levels [SD] (0.78 mg/dL [0.65 mg/dL] for ABPC, 1.2 mg/dL [0.96 mg/dL] for ABPC + CTRX, and 0.73 mg/dL [0.50 mg/dL] for ABPC/MCIPC), aspartate aminotransferase (AST) levels [SD] (42.6 U/L [81.7 U/L] for ABPC, 120 U/L [493 U/L] for ABPC + CTRX, and 54.1 U/L [68.5 U/L] for ABPC/MCIPC), and alanine aminfotransferase levels [SD] (33.1 U/L [36.0 U/L] for ABPC, 98.2 U/L [380 U/L] for ABPC + CTRX, and 46.6 U/L [67.6 U/L] for ABPC/MCIPC). Anemia and thrombocytopenia on admission were indicated based on hemoglobin levels (10.8 g/dL [2.1 g/dL] for ABPC, 11.6 g/dL [2.50 g/dL] for ABPC + CTRX, and 11.9 g/dL [2.80 g/dL] for ABPC/MCIPC) and platelet levels (204 10^3^/μL [105 10^3^/μL] for ABPC, 155 10^3^/μL [116 10^3^/μL] for ABPC + CTRX, and 256 10^3^/μL [160 10^3^/μL] for ABPC/MCIPC), respectively, showing significant differences.

Patient outcomes are presented in Table 2. Significant differences were noted in the following parameters for ABPC, ABPC + CTRX, and ABPC/MCIPC: duration of treatment (14.0 days [12.1 days], 9.3 days [8.8 days], 29.0 days [19.2 days]); duration of hospital stay (34.2 days [31.1 days], 45.8 days [47.4 days], 51.6 days [32.5 days]); proportion of patients with AKI KDIGO grade 2–3 (9.0%, 12.3%, 30.2%), grade 2–4 thrombocytopenia (10.9%, 36.8%, 16.3%), and grades 2–4 AST level elevation (10.2%, 22.8%, 20.9%); 30-day intensive care unit admission (6.5%, 42.1%, 14.0%); use of mechanical ventilation (5.4%, 24.6%, 20.9%); and the use of vasopressors (10.5%, 31.6%, 23.3%). However, there were no differences in 30-day (4.3%, 7.0%, 2.3%) and 90-day (8.3%, 14.0%, 7.0%) mortality rates.

Figure 1 shows the Kaplan–Meier analysis of the non-AKI incidence and log-rank test results for the non-AKI incidence curve. ABPC/MCIPC was associated with a significantly higher incidence of AKI than that associated with ABPC (*p* = 0.08).

We used Cox proportional hazard models with covariates including age, sex, use of mechanical ventilation, use of vasopressors, proportion of patients with qSOFA score > 2, incidence of chronic kidney disease, and duration of therapy in addition to ABPC/MCIPC to evaluate the time of the occurrence of AKI events (Table 3). ABPC/MCIPC significantly contributed to the occurrence of AKI events, with a hazard ratio (HR) of 1.83 (95% confidence interval of 1.22–2.74, *p* = 0.003).

In the propensity score (PS)-matched cohort, 21 patients treated with ABPC were matched to 21 patients treated with ABPC/MCIPC using a caliper value of 0.2. In the PS-matched cohort, the area under the receiver operating characteristic curve (AUC) was 0.932. Baseline characteristics of the PS-matched cohort are presented in Appendix A. In the PS-matched cohort, the incidence of AKI between the ABPC and ABPC/MCIPC groups was 0% (0/0) versus 23.8% (5/21), with a *p*-value of 0.048 (Appendix A). ABPC/MCIPC was associated with a significantly higher risk of AKI than that of the other groups.

## 3. Discussion

In this study, although CTRX + ABPC was relatively more frequently administered in severe cases alongside ABPC, significant adverse events were observed only in terms of thrombocytopenia and liver dysfunction, with no direct association with mortality. Furthermore, we identified a significant association between ABPC/MCIPC and an increased risk of AKI compared with ABPC. More importantly, this finding remained consistent even after adjusting for confounding factors via PS matching.

ABPC is highly effective against Gram-positive bacterial strains, particularly penicillin-sensitive *Streptococcus pneumoniae* (PSSP), beta-hemolytic *Streptococcus* spp., and *Enterococcus* spp. Guidelines for treating bacterial meningitis [4,5] and infective endocarditis [6,7] recommend double beta-lactam therapy for PSSP and *E. faecalis.* Enterobacteriaceae are often the causative agents in urinary tract infections, making CTRX effective and frequently used. However, when Gram-positive cocci in chains are found in urine, Gram staining using CTRX becomes challenging because *Enterococci* have intrinsic resistance to cephalosporins. As Enterobacteriaceae produce beta-lactamases, the use of ABPC/SBT should be considered. However, ABPC/SBT is not effective against Enterobacteriaceae in Japan [8]. Consequently, clinicians often prefer to administer piperacillin/tazobactam owing to its broad-spectrum activity against Gram-positive cocci and Gram-negative rods [9].

The combination of ABPC with CTRX shows a high potential for overcoming the issues of the antimicrobial spectrum and may significantly reduce the consumption of broad-spectrum antibiotics. To investigate the safety of ABPC + CTRX, we conducted a literature review on the combination therapy of ABPC + CTRX using the Embase, PubMed, and Central databases on 30 September 2023. Consequently, we found six studies (one randomized controlled study and five observational studies) that used ABPC + CTRX for the treatment of infective endocarditis caused by *E. faecalis*, as detailed in Table 4. This review revealed that the incidence of renal impairment ranges from 0 to 33%, whereas that of leukopenia ranges from 0 to 2.8%. Despite involving the administration of CTRX every 12 or 24 h in our study, our findings on adverse events were within the range of complications observed in the review. In this review, it was unclear whether the incidence of AKI-related complications varied across studies. A retrospective study in France examined AKI occurrence in infective endocarditis cases and reported that the risk is not associated with beta-lactam antibiotics but with conditions such as heart failure and vancomycin administration [10]. Another retrospective study in France that investigated early AKI occurring in infective endocarditis cases reported that factors such as infective endocarditis due to *S. aureus*, use of vasopressors, history of diabetes, history of peripheral artery disease, and immunological manifestations are significantly associated with AKI [11]. Based on these findings, we believe that the safety of ABPC + CTRX is comparable to that of ABPC alone.

Based on our findings, ABPC/MCIPC is not a suitable candidate for the treatment of MSSA bacteremia owing to the high risk of AKI. In Japan, only the combined formulation of ABPC/MCIPC is commercially available, necessitating combination therapy with beta-lactam antimicrobials. Despite the retrospective nature of these studies, no difference in treatment failure was found between CEZ and ASPs, although a relatively higher incidence of AKI and liver dysfunction is associated with treatment with anti-MSSA agents [12,13,14,15,16]. Recent research has shown no differences in overall treatment failure between patients with PSSP bacteremia treated with benzylpenicillin and anti-staphylococcal beta-lactam CEZ or MCIPC [17]. Thus, the use of ABPC/MCIPC for treating MSSA in Japan may be limited in the future.

Our study has a few limitations. This was a single-center retrospective cohort study, which increased the potential for bias depending on the antibiotic group. In our study, data extraction was based on the type of antibiotic administered, making it impossible to separate cases based on specific diseases. Therefore, the ABPC group may have included patients with cellulitis or necrotizing fasciitis. Similarly, the ABPC + CTRX group may have included patients with urinary tract infections, infective endocarditis, or bacterial meningitis. Furthermore, the daily dose of CTRX varied, with few patients receiving 1, 2, or 4 g CTRX/day. Consequently, the severity of illness and antibiotic dosage may vary among the antibiotic treatment groups. To mitigate these confounding factors, we extracted data from patients with bacteremia and used Cox proportional hazard models and PS matching. Despite these limitations, our results remained consistent. The AUC between 0.6 and 0.9 suggests moderate-to-good discriminative ability for a predictive model. Regarding the incidence of resistant strains, we could not follow up in this study. In several studies, the use of CTRX is associated with the risk of inducing ESBL-producing bacteria [18,19]. On the other hand, exposure to PIPC/TAZ may lead to PIPC/TAZ resistance, necessitating the use of broader-spectrum antibiotics such as carbapenems and cefiderocol. Further prospective studies are necessary to thoroughly assess the safety of ABPC + CTRX, and the findings from such studies will play a pivotal role in reducing the reliance on broad-spectrum antimicrobial agents.

**Table 4 antibiotics-13-00696-t004:** Literature review of the study using ABPC plus CTRX.

1	A. Ramos-Martinez, et al., 2020 [20]/non-randomized prospective cohort study/*E. faecalis* NVE		ABPC 2 g q4h plus CTRX 2 g q12h for 4 weeks (39)	2 g q4h plus CTRX 2 g q12h for 6 weeks (70)
		Previous renal failure	11/39(28.2%)	17/70(24.3%)
		Renal impairment/glomerulonephritis	10/39(25.6%)/0/39(0%)	20/70(28.6%)/1/39(3%)
		Leukopenia	0/39(0%)	2/70(2.8%)
		In-hospital mortality	4/39(10.3%)	8/70(11.4%)
2	A. El Rafei, et al., 2018 [21]/retrospective cohort study/*E. faecalis* NVE, PVE		ABPC 2 g q4h plus CTRX 2 g q12h for 4–6 weeks (18)	ABPC 2 g q4h plus GM 3 mg/kg/day for 4–6 weeks (67)
		AKI	2/18(11%)	17/67(25%)
		Leukopenia	0/18(0%)	0/67(0%)
		1-year mortality	2/18(11%)	6/67(9%)
			ABPC + CTRX	
3	J. M. Pericas, et al., 2018 [22]/retrospective analysis of a prospective collected data/*E. faecalis* NVE, PVE		ABPC 2 g q4h plus CTRX 2 g q12h for 4–6 weeks (46)	ABPC 2 g q4h plus GM 3 mg/kg/day for 4–6 weeks (32)
		CKD/HD	12/46(26%)/1/46(2%)	8/32(25%)/3/32(9%)
		AKI	15/46(33%)	20/32(63%)
		myelotoxicity	1/46(2%)	0/32(0%)
		1-year mortality	11/46(24%)	10/32(31%)
4	N. H. Shah, et al., 2021 [23]/propensity score-matched retrospective cohort study/*E. faecalis* NVE, PVE		ABPC plus CTRX(100)	ABPC plus GM(90)
		CKD/HD	NP	NP
		AKI	6/100(6%)	13/90(14%)
		Leukopenia	1/100(1%)	4/90(4%)
		90-day mortality	6/56(10.7%) after PS	4/56(7.1%) after PS
5	J. Gavalda, et al., 2007 [24]/observational, open-label, non-randomized cohort study/*E. faecalis* NVE, PVE		ABPC 2 g q4h plus CTRX 2 g q12h for 6 weeks (43)	
		CKD	8/43(19%)	
		AKI	0/43(0%)	
		Leukopenia	1/43(2%)	
		All-cause mortality	12/43(28%)	
6	N. Fernández-Hidalgo, et al., 2013 [25]/non-randomized, non-blinded cohort study/*E. faecalis* NVE, PVE		ABPC 2 g q4h plus CTRX 2 g q12h for 4–6 weeks (159)	ABPC 2 g q4h plus GM 3 mg/kg/day for 4–6 weeks (87)
		CKD/HD	53/159(33%)/12/159(8%)	14/87(16%)/3/87(3%)
		AKI	53/159(33%)	40/87(46%)
		Leukopenia	1%(only treatment interruption)	0%
		All-cause mortality	42/159(26%)	22/87(25%)

Abbreviations: *E. faecalis*, *Enterococcus faecalis*; PVE, prosthetic valve endocarditis; NVE, native valve endocarditis; AKI, acute kidney injury; CKD, chronic kidney injury; HD, hemodialysis; ABPC, ampicillin; CTRX, ceftriaxone; GM, gentamycin.

## 4. Materials and Methods

### 4.1. Study Design and Setting

This single-center retrospective cohort study was performed at St. Luke’s International Hospital, a 520-bed teaching hospital in Tokyo, Japan.

### 4.2. Ethics and Informed Consent

This study was conducted according to the guidelines of the Declaration of Helsinki and was approved by the Institutional Review Board of St. Luke’s International Hospital in Tokyo, Japan (22-R049). The requirement for informed consent was waived due to the retrospective nature of the study.

### 4.3. Inclusion and Exclusion Criteria

We included adult patients (aged ≥ 18 years) with bloodstream infections who received one of three antibiotic treatments for >3 days: ABPC, ABPC + CTRX, or ABPC/MCIPC between 2004 and 2022. We included antimicrobials used for both empirical and optimal treatment therapies. We also considered non-beta-lactam antibiotics and other non-antimicrobial drugs that had been administered for ≥3 days in combination with beta-lactams. Eligible patients were identified by screening an electronic hospital database. Exclusion criteria were as follows: patients on hemodialysis, individuals with missing data, those who had taken more than three beta-lactam antimicrobials, or those treated with beta-lactam antibiotics other than ABPC, CTRX, or ABPC/MCIPC.

### 4.4. Study Outcomes

The primary outcome of this study was the incidence of adverse events, such as acute renal injury, liver dysfunction, and myelosuppression, between patients treated with ABPC, ABPC + CTRX, and ABPC/MCIPC. The severity of acute kidney injury (AKI) was classified according to the KDIGO classification [26]. Other adverse events were defined according to the Common Terminology Criteria for Adverse Events v5.0. Other adverse events included anemia, leukopenia, thrombocytopenia, hepatic dysfunction, use of mechanical ventilation, use of vasopressors, and 30/90-day mortality.

### 4.5. Statistical Analysis

Bivariate analysis used the Chi-squared and Fisher’s exact tests for categorical variables and the *t*-test for continuous variables. Kaplan–Meier curves were used to illustrate the incidence of non-AKI based on the types of antibiotics administered. Differences in the non-AKI incidence curves between groups were compared using the log-rank test. Where significant differences existed between antibiotics in the log-rank test, the Cox proportional hazards model was applied to estimate the HRs. We applied the forced entry method. To control for potential confounders in assessing the risk of AKI, we conducted PS matching based on the baseline characteristics between the groups. The variables used for this matching included all factors associated with AKI, such as patient demographic characteristics, medications, comorbidities, laboratory values, and patient severity because these are designed to generate good predictive models of exposure [27]. SPSS uses the forced entry method for variable selection. We used a caliper value of 0.2 to perform PS matching and then estimated the AUC from the matched PS. The standardized mean difference was calculated using Cohen’s d for the *t*-test and phi for the Chi-squared test. All analyses were conducted using the SPSS 19.0 J statistical software (IBM Japan, Tokyo, Japan).

## 5. Conclusions

Empirical treatment with a double beta-lactam regimen (ABPC + CTRX) offers broad coverage against Gram-positive cocci (*Streptococcus* spp., MSSA, and *E. faecalis)* and a wide range of Enterobacteriaceae, thereby reducing the need for broader-spectrum beta-lactam/beta-lactamase inhibitors and carbapenems. ABPC/MCIPC is not recommended for treating MSSA bacteremia owing to the elevated risk of AKI. ABPC + CTRX may be safe, whereas ABPC/MCIPC presents a higher risk of AKI and may not be suitable.

## Figures and Tables

**Figure 1 antibiotics-13-00696-f001:**
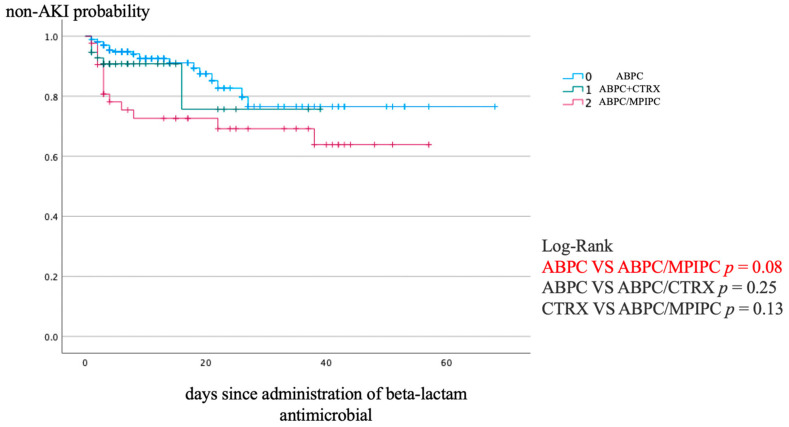
Kaplan–Meier analysis of non-AKI incidence and the log-rank test result for the non-AKI incidence curve. AKI, acute kidney injury.

**Table 1 antibiotics-13-00696-t001:** Baseline characteristics of patients with bacteremia.

*n*	ABPC(277)	ABPC plus CTRX(57)	ABPC/MCIPC(43)	*p*
age(SD) (years)	71.1(16.5)	64.0(16.9)	56.6(17.5)	<0.01
female(%)	173(62.5)	27(47.4)	31(72.1)	0.031
congestive heart failure(%)	69(24.9)	9(15.8)	11(25.6)	0.32
diabetes(%)	87(31.4)	20(35.1)	7(16.3)	0.091
respiratory disease(%)	135(48.7)	29(50.9)	17(39.5)	0.48
myocadiac infarction(%)	24(8.7)	1(1.8)	2(4.7)	0.15
collagen disease(%)	4(1.4)	1(1.8)	2(4.7)	0.35
liver dysfunction(%)	30(10.8)	10(17.5)	4(9.3)	0.31
cancer(%)	110(39.7)	17(29.8)	6(14.0)	0.003
chronic kidney disease(%)	108(39.0)	20(35.1)	8(18.6)	0.034
cerebrovascular(%)	35(12.6)	9(15.3)	6(14.0)	0.81
hypertension(%)	150(54.2)	20(35.1)	20(46.5)	0.028
HIV(%)	0(0)	0(0)	2(4.7)	<0.01
qSofa ≥ 2(%)	63(23.1)	38(64.4)	18(41.9)	<0.01
CLDM(%)	53(19.1)	14(24.6)	19(44.2)	0.001
VCM(%)	21(7.6)	16(28.1)	18(41.9)	<0.01
AG(%)	37(13.4)	4(7.0)	4(9.3)	0.35
L-AMPHB(%)	0(0)	0(0)	1(2.3)	0.02
ACV(%)	1(0.4)	5(8.8)	0(0)	<0.01
NSAIDs(%)	59(21.3)	5(8.8)	10(23.3)	0.078
ACE inhibitor(%)	18(6.5)	3(5.3)	3(7.0)	0.92
ARB(%)	34(12.3)	5(8.8)	10(23.3)	0.081
ACE inhibitor/ARB(%)	50(18.1)	8(14.0)	12(27.9)	0.19
diuretics(%)	49(17.7)	4(7.0)	10(23.3)	0.069
chemotherapy(%)	6(2.2)	0(0)	0(0)	0.34
calcineurin inhibitor(%)	0(0)	1(1.8)	0(0)	0.06
contrast CT(%)	75(27.1)	28(49.1)	26(60.5)	<0.001
on admission, eGFR(SD) (mL/min/1.73 m^2^)	78.8(46.3)	80.8(43.0)	102.1(54.6)	0.011
T-bil(SD) (mg/dL)	0.78(0.65)	1.2(0.96)	0.73(0.50)	<0.001
ALP(SD) (U/L)	314.8(271.0)	370.2(245.7)	328.9(243.7)	0.4
AST(SD) (U/L)	42.6(81.7)	120.4(493.0)	54.1(68.5)	0.035
ALT(SD) (U/L)	33.1(36.0)	98.2(380.0)	46.6(67.6)	0.014
GTP(SD) (U/L)	76.2(107.6)	101.1(135.8)	75.5(70.5)	0.47
CRP(SD) (mg/dL)	10.9(9.3)	15.3(12.3)	13.5(10.1)	0.005
WBC(SD) (10^3^/μL)	10.9(6.1)	11.9(8.9)	11.3(5.1)	0.53
Hgb(SD) (g/dL)	10.8(2.1)	11.6(2.5)	11.9(2.8)	0.002
PLT(SD) (10^3^/μL)	203.6(105.0)	154.7(116.0)	255.9(160.0)	<0.001
duration of treatment(SD) (day)	14.0(12.1)	9.3(8.8)	29.0(19.2)	<0.01
use of mechanical ventilation(%)	15(5.4)	14(24.6)	9(20.9)	<0.001
use of vasopressor(%)	29(10.5)	18(31.6)	10(23.3)	<0.001

Abbreviations: SD, standard deviation; HIV, human immunodeficiency virus; CLDM, clindamycin; VCM, vancomycin; AG, aminoglycoside; L-AMPHB, liposomal-amphotericin B; ACV, acyclovir; NSAIDs, non-steroidal anti-inflammatory drugs; ACE, angiotensin-converting enzyme; ARB, angiotensin II receptor blocker; CT, computed tomography; eGFR, estimated glomerular filtration rate; ALP, alkaline phosphatase; AST, aspartate aminotransferase; ALT, alanine aminotransferase; GTP, guanosine triphosphate; CRP, C-related protein; WBC, white blood cell count; Hgb, hemoglobin; PLT, platelate.

**Table 2 antibiotics-13-00696-t002:** Patients’ outcomes.

*n*	ABPC(277)	ABPC plus CTRX(57)	ABPC/MCIPC(43)	*p*
Length of stay(SD) (day)	34.2(31.1)	45.8(47.4)	51.6(32.5)	0.03
AKI KDIGO Grade 2–3(%)	25(9.0)	7(12.3)	13(30.2)	<0.001
Stage 1(%)	18(6.5)	3(5.3)	5(11.6)	
Stage 2(%)	4(1.4)	1(1.8)	3(7.0)	
Stage 3(%)	3(1.1)	3(5.3)	5(11.6)	
Grade 2–4 leukopenia(%)	21(7.6)	7(12.3)	2(4.7)	0.35
Grade 2(%)	11(4.0)	4(7.0)	2(4.7)	
Grade 3(%)	7(2.5)	2(3.5)	0(0)	
Grade 4(%)	3(1.1)	1(1.8)	0(0)	
Grade 2–4 anemia(%)	143(51.8)	29(50.9)	23(53.5)	0.97
Grade 2(%)	103(37.3)	16(28.1)	13(30.2)	
Grade 3(%)	31(11.2)	10(17.5)	7(16.3)	
Grade 4(%)	9(3.3)	3(5.3)	3(7.0)	
Grade 2–4 thrombocytopenia(%)	30(10.9)	21(36.8)	7(16.3)	<0.001
Grade 2(%)	8(2.9)	6(10.5)	5(11.6)	
Grade 3(%)	12(4.3)	6(10.5)	1(2.3)	
Grade 4(%)	10(3.6)	9(15.8)	1(2.3)	
Grade 2–4 T-bil elevation(%)	13(5.1)	5(8.9)	3(7.1)	0.6
Grade 2(%)	12(4.7)	2(3.6)	3(7.1)	
Grade 3(%)	1(0.4)	3(5.4)	0(0)	
Grade 4(%)	0(0)	0(0)	0(0)	
Grade 2–4 ALP elevation(%)	17(8.7)	5(9.6)	4(9.8)	0.97
Grade 2(%)	14(7.2)	5(9.6)	4(9.8)	
Grade 3(%)	3(1.5)	0(0)	0(0)	
Grade 4(%)	0(0)	0(0)	0(0)	
Grade 2–4 G-GTP elevation(%)	55(24.6)	19(37.3)	12(28.6)	0.21
Grade 2(%)	33(14.7)	12(23.5)	5(11.9)	
Grade 3(%)	22(9.8)	7(13.7)	7(16.7)	
Grade 4(%)	0(0)	0(0)	0(0)	
Grade 2–4 AST elevation(%)	28(10.2)	13(22.8)	9(20.9)	0.014
Grade 2(%)	13(4.7)	8(14.0)	2(4.7)	
Grade 3(%)	13(4.7)	3(5.3)	7(16.3)	
Grade 4(%)	2(0.7)	2(3.5)	0(0)	
Grade 2–4 ALT elevation(%)	26(9.5)	11(19.3)	7(16.3)	0.07
Grade 2(%)	16(5.8)	6(10.5)	3(7.0)	
Grade 3(%)	10(3.6)	3(5.3)	4(9.3)	
Grade 4(%)	0(0)	2(3.5)	0(0)	
30-day mortality(%)	12(4.3)	4(7.0)	1(2.3)	0.51
90-day mortality(%)	23(8.3)	8(14.0)	3(7.0)	0.35
30-day ICU admission(%)	18(6.5)	24(42.1)	6(14.0)	<0.001

Abbreviations: SD, standard deviation; AKI, acute kidney injury; ALP, alkaline phosphatase; AST, aspartate aminotransferase; ALT, alanine aminotransferase; GTP, guanosine triphosphate; ICU, intensive care unit.

**Table 3 antibiotics-13-00696-t003:** Cox hazard analysis for AKI events.

	Hazard Ratio	95% CI	*p*
Age	0.992	0.971–1.01	0.474
sex (male:0, female:1)	1.04	0.469–2.29	0.929
use of mechanical ventilation	1.79	0.591–5.42	0.303
ABPC/MCIPC (ABPC:0, ABPC/MCIPC:2)	1.83	1.22–2.74	0.003
vasopressor	3.52	1.22–10.1	0.020
qSOFA more than 2	0.493	0.189–1.29	0.148
chronic kidney disease	1.99	0.907–4.35	0.086
duration of therapy	0.976	0.950–1.00	0.088

Abbreviations: CI, confidence interval; AKI, acute kidney injury; ABPC, ampicillin; MCIPC, cloxacillin.

## Data Availability

The data presented in this study are available from the corresponding author (K.I.) upon reasonable request.

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
