# Peer review of "Adverse Events Comparison of Double Beta-Lactam Combinations for Bloodstream Infections: Ampicillin plus Ceftriaxone and Ampicillin/Cloxacillin"

_antibiotics, 2024, doi:10.3390/antibiotics13080696_

Round 1

Reviewer 1 Report

Comments and Suggestions for Authors

This manuscript aimed to compare the incidence of adverse events, such as acute renal injury, liver dysfunction, and myelosuppression, between patients treated with ABPC, ABPC+CTRX, and ABPC/MCIPC. The study found that ABPC+CTRX treatment may be safe under certain conditions, whereas ABPC/MCIPC treatment may present a higher risk of developing acute kidney injury (AKI) and not be suitable for treating bacterial infections. This is a well-written manuscript, the author made a great effort, but I suggest some changes that must be made to the text, to contribute to a better understanding of the points they are trying to make.

comments

1.     In line 68, author used 277 patients who received ABPC, 57 who received ABPC+CTRX, and 43 who received ABPC/MCIPC. Huge difference in number for comparison. Please explain.

2.      Is AKI related to age and sex. Please explain.

Author Response

Dear Reviewer 1,

Thank you very much for your excellent feedback. The constructive discussion has significantly advanced the arguments in our paper. We have written a response letter addressing your comments, which we kindly ask you to review.

Best regards,

Kazuhiro Ishikawa

#Reviewer 1

This manuscript aimed to compare the incidence of adverse events, such as acute renal injury, liver dysfunction, and myelosuppression, between patients treated with ABPC, ABPC+CTRX, and ABPC/MCIPC. The study found that ABPC+CTRX treatment may be safe under certain conditions, whereas ABPC/MCIPC treatment may present a higher risk of developing acute kidney injury (AKI) and not be suitable for treating bacterial infections. This is a well-written manuscript, the author made a great effort, but I suggest some changes that must be made to the text, to contribute to a better understanding of the points they are trying to make.

comments

  1. In line 68, author used 277 patients who received ABPC, 57 who received ABPC+CTRX, and 43 who received ABPC/MCIPC. Huge difference in number for comparison. Please explain.

Response>Thank you for feedback. In our cases, most patients treated with ABPC have cellulitis or urinary tract infections caused by Streptococcus species. Most patients treated with ABPC + CTRX have bacterial meningitis covered by Listeria monocytogenes and Streptococcus pneumoniae, and infective endocarditis caused by Enterococcus faecalis. Additionally, ABPC/MPIPC is used for bacteremia, meningitis, and epidural or brain abscesses caused by MSSA.

  1. Is AKI related to age and sex. Please explain.

   Response> Thank you for your feedback. Generally, elderly and male patients are reported to be at risk for AKI (PMID: 25943717). However, in our study, AKI was not found to be a risk factor. Possible reasons include differences in the sample population since our cohort consisted of patients with bacteremia, or the sample size was not large enough to detect a significant difference.

Reviewer 2 Report

Comments and Suggestions for Authors

The authors planned the work effectively to achieve the aim and discussed the results neatly with adequate statistical validations. 

Minor comments: 

1. What do author mean by this “ABPC/MCIPC” group of test case. Administered with both antibiotics on alternative conditions or either one. If it's either one, what is the significance in this group compared to “ABPC” group. Information about this would be ideal for the significance of each test cases. 

2. Based on the data, authors depict ABPC+CTRX is effective and can be used as broad coverage antibiotics. However, the probability of acquiring resistance is not discussed. 

3. The mechanism/mode of action and administration procedure are vital in mitigating the bacterial infection. Are the test cases used in the study used the same method of antibiotic administration. If varied, how does it affect the sample homogeneity.

Author Response

Dear Reviewer 2,

Thank you very much for your excellent feedback. The constructive discussion has significantly advanced the arguments in our paper. We have written a response letter addressing your comments, which we kindly ask you to review.

Best regards,

Kazuhiro Ishikawa

#

#Reviwer 2

Comments and Suggestions for Authors

The authors planned the work effectively to achieve the aim and discussed the results neatly with adequate statistical validations. 

Minor comments: 

  1. What do author mean by this “ABPC/MCIPC” group of test case. Administered with both antibiotics on alternative conditions or either one. If it's either one, what is the significance in this group compared to “ABPC” group. Information about this would be ideal for the significance of each test cases.

Response>Thank you for pointing this out. The "ABPC/MCIPC" group refers to the Viccillin-S® (Ampicillin Sodium,Cloxacillin Sodium Hydrate) sold in Japan. In Japan, MCIPC is not available as a single agent. In our cases, ABPC/MPIPC was used for mainly MSSA infections.

  1. Based on the data, authors depict ABPC+CTRX is effective and can be used as broad coverage antibiotics. However, the probability of acquiring resistance is not discussed.

Response> Thank you for your response. ABPC+CTRX has a narrower spectrum compared to PIPC/TAZ but provides broader coverage against Enterobacteriaceae compared to ABPC/SBT. The use of CTRX is associated with the risk of inducing ESBL-producing bacteria (PMID: 30478179, PMID: 25694654). On the other hand, exposure to PIPC/TAZ may lead to PIPC/TAZ resistance, necessitating the use of broader spectrum antibiotics such as carbapenems and cefiderocol. In our study, we were unable to follow up on the emergence of resistant strains. Therefore, a prospective study on ABPC+CTRX is needed. I added this sentence in the discussion parts.

  1. The mechanism/mode of action and administration procedure are vital in mitigating the bacterial infection. Are the test cases used in the study used the same method of antibiotic administration. If varied, how does it affect the sample homogeneity.

Response>Thank you for your response. For ABPC, our hospital protocol recommends 2g every 4 hours. For CTRX, the protocol recommends 1g-2g every 24 hours for urinary tract infections, or 2g every 12 hours for bacterial meningitis and infective endocarditis. For ABPC/MCIPC, the protocol recommends 4g every 6 hours. Additionally, the dosage is adjusted based on renal function. Since this study based on a retrospective cohort study, the dosage vary depending on the clinician. To mitigate the bias in this study due to differences in severity and dosage, we selected patients with bacteremia as the study population.

Reviewer 3 Report

Comments and Suggestions for Authors

Research pursued by Ishikawa et al. provides insights into the safety profiles of different combinations of beta-lactam antibiotics for the treatment of bacterial infections. Groups of patients treated with ampicillin (ABPC), ampicillin+ceftriaxone (ABPC+CTRX), and ampicillin+cloxacillin (ABPC/MCIPC) were compared.

                The authors concluded from their observations that ABPC/MCIPC treatment, which is the only combination therapy available in Japan for the treatment of S. aureus bacteremia, may present a higher risk for the development of acute kidney injury. Additionally, their findings suggest that ABPC+CTRX therapy may be safely administered to the patients under certain conditions.

The authors note that the study design has inherent biases. Moreover, this study was conducted at a single hospital in Japan, which may not generalize the findings to other healthcare settings within or outside the country. However, such results are important and may become the basis for decision making among infectious disease experts in Japan. It is for this reason, I recommend publishing the paper. Following are my comments:

1)      L56-59: Here, authors discuss that “the use of double beta-lactams may reduce the reliance on broad spectrum antibiotics”, after the discussion regarding the emergence of antibiotic resistance. Will the use of double beta-lactams not give bacteria a chance to develop antibiotic resistance, as a single mechanism may be enough to inhibit both the antibiotics in the treatment? If this could be true, then the use of beta-lactam+beta-lactamase inhibitors is actually better in general?

2)      The standard deviations are huge. Authors may obtain better results with the inclusion of larger groups of patients.

3)      Tables constitute a major proportion of the manuscript. It would be easier for the readers if the authors convert the tabulated data into some form of plots/figures.

Author Response

Dear Reviewer 4,

Thank you very much for your excellent feedback. The constructive discussion has significantly advanced the arguments in our paper. We have written a response letter addressing your comments, which we kindly ask you to review.

Best regards,

Kazuhiro Ishikawa

#Reviwer 3

Comments and Suggestions for Authors

Research pursued by Ishikawa et al. provides insights into the safety profiles of different combinations of beta-lactam antibiotics for the treatment of bacterial infections. Groups of patients treated with ampicillin (ABPC), ampicillin+ceftriaxone (ABPC+CTRX), and ampicillin+cloxacillin (ABPC/MCIPC) were compared.

                The authors concluded from their observations that ABPC/MCIPC treatment, which is the only combination therapy available in Japan for the treatment of S. aureus bacteremia, may present a higher risk for the development of acute kidney injury. Additionally, their findings suggest that ABPC+CTRX therapy may be safely administered to the patients under certain conditions.

The authors note that the study design has inherent biases. Moreover, this study was conducted at a single hospital in Japan, which may not generalize the findings to other healthcare settings within or outside the country. However, such results are important and may become the basis for decision making among infectious disease experts in Japan. It is for this reason, I recommend publishing the paper. Following are my comments:

1)      L56-59: Here, authors discuss that “the use of double beta-lactams may reduce the reliance on broad spectrum antibiotics”, after the discussion regarding the emergence of antibiotic resistance. Will the use of double beta-lactams not give bacteria a chance to develop antibiotic resistance, as a single mechanism may be enough to inhibit both the antibiotics in the treatment? If this could be true, then the use of beta-lactam+beta-lactamase inhibitors is actually better in general?

Response> Thank you for your feedback. We could not find evidence of resistance development through long-term follow-up or systematic review of infectious endocarditis regarding the use of double beta-lactams. The extent to which resistance mechanisms increase compared to ampicillin alone remains a topic for future discussion. Reviewer 2 raised similar points, noting that the use of third-generation cephalosporins increases the risk of ESBL-producing bacteria. On the other hand, reducing the use of antibiotics such as PIPC/TAZ can decrease the incidence of PIPC/TAZ-resistant Enterobacteriaceae, potentially reducing the need for broader spectrum antibiotics like carbapenems and cefiderocol. I have added these points to the Discussion section.

2)      The standard deviations are huge. Authors may obtain better results with the inclusion of larger groups of patients.

Response> Thank you for your feedback. In this retrospective cohort study, the number of patients was limited. Further multicenter studies are needed.

3)      Tables constitute a major proportion of the manuscript. It would be easier for the readers if the authors convert the tabulated data into some form of plots/figures.

Response> Thank you for your feedback. I will discuss the layout of the table with the editor.

Reviewer 4 Report

Comments and Suggestions for Authors

Please have a look at the attached file.

Comments on the Quality of English Language

Author Response

Dear Reviewer 4,

Thank you very much for your excellent feedback. The constructive discussion has significantly advanced the arguments in our paper. We have written a response letter addressing your comments, which we kindly ask you to review.

Best regards,

Kazuhiro Ishikawa

#Reviwer4

This paper discusses the adverse events of double beta-lactam administration in patients with bacteremia. The authors discuss well. However, to make the article more understandable, the following sections of the manuscript must be addressed.

  1. I recommend the authors to keep their heading as simple and legible as possible. Authors can revise on their own. An example can be “Adverse events comparison in double beta-lactam combinations treatment for bloodstream infections: Ampicillin plus ceftriaxone versus ampicillin/cloxacillin”

Response>Thank you for feedback. I modified the title.

  1. In line 18, I advise that the authors provide instances of what they mean by adverse events, such as acute renal damage, liver dysfunction, and myelosuppression.

Response>Thank you for feedback. I added the “such as acute kidney injury, liver dysfunction, and myelosuppression in line 19”.

  1. In line 26, the authors reference AMPC. Is this a mistake or anything else?

Response>Thank you for feedback. I modified the ABPC/MCIPC.

  1. In line 27, it is better to mention the specific conditions rather than broadening them.

Response>Thank you for feedback. I removed the “under certain conditions”.

ABPC+CTRX may be safe under certain conditions, whereas ABPC/MCIPC presents a higher risk of AKI and may not be suitable.

  1. The authors have not mentioned whether the bloodstream infections they are reporting are due to Staphylococcus aureus or if there are additional cases of bacteremia caused by MSSA and E. faecalis. This part needs to be clear in the material and method section.

Response> Thank you for your feedback. In this study, we could not extract the strains in the positive blood cultures. Therefore, this is a limitation of our study.

  1. Why did the authors choose the ABPC + CTRX combination? Is this the sole combination for treating bacteremia in Japan?

Response> Thank you for your feedback. We chose ABPC+CTRX because, in situations where Enterobacteriaceae show decreased susceptibility to ABPC/SBT, there is a high likelihood of choosing PIPC/TAZ, thus reducing its usage frequency. While cefotaxime could also be chosen as a third-generation cephalosporin, its usage frequency is low, and combining it with ABPC would increase the number of doses required. Therefore, we selected ABPC+CTRX for its larger sample size and greater convenience due to fewer doses.

  1. Since this is a retrospective study, how do the authors account for potential bias and validity of the results?

Response> Thank you for your feedback. As mentioned in the limitation, ABPC is used for streptococcus species bacteremia, ABPC+CTRX for E. faecalis infective endocarditis or bacterial meningitis, and ABPC/MPIPC for S. aureus bacteremia. Therefore, there might be a selection bias based on the disease.

  1. Which factors, and for what reason, were part of the propensity score model? Were any potential confounders overlooked?

Response> Thank you for your feedback. The matched factors were included in all baseline factors in Supplementary Table 1.

  1. In line 185, please verify whether it is AMPC or ABPC.

Response>Thank you for feedback. I modified ABPC.

  1. English language and style are acceptable, although a revision is needed.

Response> Thank you for your feedback. Our manuscript was checked by the proofreading service again.

#Reviwer5

Comments and Suggestions for Authors

The authors should consider the followings:

The authors should explain why they use such a wide coverage of primary outcomes seems, would the authors narrow down the primary outcome? Any secondary outcome the authors designed?

Response> Thank you for your feedback. Our aim in this study was to investigate the adverse events of the double beta-lactam. Initially, we also considered including treatment outcomes, but we excluded them because the diseases were different.

With the covid pandemic, or other major flu seasons or pandemic diseases in japan, did the authors considering these in their period between 2004 and 2022? If yes, what measures did the authors perform to exclude the pandemic factors, which may interfere with the study target(s)?

Response> Thank you for your feedback. The number of patients hospitalized for infectious diseases increased due to COVID-19, which may have influenced other infectious disease patients' outpatient visits to some extent. However, this study focused on bacteremia patients, and since hospitalization is necessary for this condition, we believe the impact is minimal.

The authors should enrich to cover the most relevant and updated references of others' works. The recall bias and observer bias should be well explained in the current study. The analytical protocol of the missing data pipeline should be well described and justified.

Response> Thank you for your feedback. This systematic review includes the latest studies, and we will continue to add new literature as it becomes available. Additionally, this is a retrospective observational study, and we have extracted data on age, gender, comorbidities, and laboratory values through chart reviews. While it may not completely eliminate early bias and observational bias, we believe these biases are minimal. Missing data from blood tests and other variables have been treated as missing values.

Why only a single-center was used in this study? Did the study results match those studies of the other center within thr greater tokyo area? Or within the asian region? (Given the same period any criteria selected)

Response> Thank you for your feedback. This study was conducted at a single feasible institution, but we aim to encourage other hospitals in Japan to participate in future retrospective multicenter studies. Additionally, this research has not been conducted in Japan before, and adverse events were not the primary focus of this study. Therefore, in the systematic review, we collected and analyzed studies on the use of double beta-lactams in infective endocarditis.

" To control for potential confounders in assessing the risk of AKI, we conducted PS matching based on the baseline characteristics between the groups".  The authors should assess the risk of bias and listed in Table 4, regarding thoss studies using ABPC plus CTRX.

Response>Thank you for feedback. I got same feedback from reviewer #4 in 7. This study has the selection bias for the disease. ABPC is used for streptococcus species bacteremia, ABPC+CTRX for E. faecalis infective endocarditis or bacterial meningitis, and ABPC/MPIPC for S. aureus bacteremia. Therefore, there might be a selection bias based on the disease.

The authors should list the study limitations of the current study.

Response> Thank you for feedback. I modified the limitation in the discussion part.

The authors should clearly present the novelty if the study in abstract and conclusion part.

Response>Thank you for feedback. The novelty of this study is written “ABPC+CTRX may be safe, whereas ABPC/MCIPC presents a higher risk of AKI and may not be suitable.” In abstract and conclusion.

Reviewer 5 Report

Comments and Suggestions for Authors

The authors should consider the followings:

The authors should explain why they use such a wide coverage of primary outcomes seems, would the authors narrow down the primary outcome? Any secondary outcome the authors designed?

With the covid pandemic, or other major flu seasons or pandemic diseases in japan, did the authors considering these in their period between 2004 and 2022? If yes, what measures did the authors perform to exclude the pandemic factors, which may interfere with the study target(s)?

The authors should enrich to cover the most relevant and updated references of others' works.

The recall bias and observer bias should be well explained in the current study.

The analytical protocol of the missing data pipeline should be well described and justified.

Why only a single-center was used in this study? Did the study results match those studies of the other center within thr greater tokyo area? Or within the asian region? (Given the same period any criteria selected)

" To control for potential confounders in assessing the risk of AKI, we conducted PS matching based on the baseline characteristics between the groups" 

The authos should assess the risk of bias and listed in Table 4, regarding thoss studies using ABPC plus CTRX.

The authors should list the study limitations of the current study.

The authors should clearly present the novelty if the study in abstract and conclusion part.

Comments on the Quality of English Language

moderate english editing required.

Author Response

Dear Reviewer 5,

Thank you very much for your excellent feedback. The constructive discussion has significantly advanced the arguments in our paper. We have written a response letter addressing your comments, which we kindly ask you to review.

Best regards,

Kazuhiro Ishikawa

#Reviwer5

Comments and Suggestions for Authors
The authors should consider the followings:

The authors should explain why they use such a wide coverage of primary outcomes seems, would the authors narrow down the primary outcome? Any secondary outcome the authors designed?

Response> Thank you for your feedback. Our aim in this study was to investigate the adverse events of the double beta-lactam. Initially, we also considered including treatment outcomes, but we excluded them because the diseases were different.

With the covid pandemic, or other major flu seasons or pandemic diseases in japan, did the authors considering these in their period between 2004 and 2022? If yes, what measures did the authors perform to exclude the pandemic factors, which may interfere with the study target(s)?

Response> Thank you for your feedback. The number of patients hospitalized for infectious diseases increased due to COVID-19, which may have influenced other infectious disease patients' outpatient visits to some extent. However, this study focused on bacteremia patients, and since hospitalization is necessary for this condition, we believe the impact is minimal.

The authors should enrich to cover the most relevant and updated references of others' works. The recall bias and observer bias should be well explained in the current study. The analytical protocol of the missing data pipeline should be well described and justified.

Response> Thank you for your feedback. This systematic review includes the latest studies, and we will continue to add new literature as it becomes available. Additionally, this is a retrospective observational study, and we have extracted data on age, gender, comorbidities, and laboratory values through chart reviews. While it may not completely eliminate early bias and observational bias, we believe these biases are minimal. Missing data from blood tests and other variables have been treated as missing values.

Why only a single-center was used in this study? Did the study results match those studies of the other center within thr greater tokyo area? Or within the asian region? (Given the same period any criteria selected)

Response> Thank you for your feedback. This study was conducted at a single feasible institution, but we aim to encourage other hospitals in Japan to participate in future retrospective multicenter studies. Additionally, this research has not been conducted in Japan before, and adverse events were not the primary focus of this study. Therefore, in the systematic review, we collected and analyzed studies on the use of double beta-lactams in infective endocarditis.

" To control for potential confounders in assessing the risk of AKI, we conducted PS matching based on the baseline characteristics between the groups".  The authors should assess the risk of bias and listed in Table 4, regarding thoss studies using ABPC plus CTRX.

Response>Thank you for feedback. I got same feedback from reviewer #4 in 7. This study has the selection bias for the disease. ABPC is used for streptococcus species bacteremia, ABPC+CTRX for E. faecalis infective endocarditis or bacterial meningitis, and ABPC/MPIPC for S. aureus bacteremia. Therefore, there might be a selection bias based on the disease.

The authors should list the study limitations of the current study.

Response> Thank you for feedback. I modified the limitation in the discussion part.

The authors should clearly present the novelty if the study in abstract and conclusion part.

Response>Thank you for feedback. The novelty of this study is written “ABPC+CTRX may be safe, whereas ABPC/MCIPC presents a higher risk of AKI and may not be suitable.” In abstract and conclusion.

Round 2

Reviewer 4 Report

Comments and Suggestions for Authors

The authors have acknowledged the comments well.

Remove 'of' from the heading.

Comments on the Quality of English Language

Looks good to me.